# Experimental Study on the Effect of Porous Media on the Aerodynamic Performance of Airfoils

**Wenjie Kong** [1], **Hao Dong** [1,2,*], **Jie Wu** [1,2], **Yidi Zhao** [1] **and Zhou Jin** [1]

[1] College of Aerospace Engineering, Nanjing University of Aeronautics and Astronautics, Nanjing 210016, China

[2] State Key Laboratory of Mechanics and Control of Mechanical Structures, Nanjing 210016, China

\* Correspondence: donghao@nuaa.edu.cn

**Abstract:** Porous media has potential applications in fluid machinery and in aerospace science and engineering due to its excellent drag-reduction properties. We carried out experimental time-resolved particle image velocimetry (TR-PIV) research, laying porous media with different pore densities on the suction side of an airfoil in the low-turbulence recirculation wind tunnel of Nanjing University of Aeronautics and Astronautics to study the effects and mechanisms of porous media on airfoil aerodynamic performance. We also used a smooth airfoil model in the experiment for comparison. Comparing the aerodynamic forces, pressure distributions, and the airfoil's suction side flow field, we found that the porous media with different pore densities had different effects on the airfoil's aerodynamic performance. Although the porous media with 20PPI (pores per inch) increased the pressure drag and reduced the airfoil lift, it considerably reduced the friction drag, thus significantly improving the airfoil's aerodynamic force. The flow visualization results indicated that, although the porous media with 20PPI reduced the circulation of flow velocity around the suction side of airfoil, it also destroyed the vortex structure, broke the low-frequency large-scale vortex into a high-frequency granular vortex, inhibited the amplitude of vortex fluctuation, reduced the shear stress on the airfoil surface, weakened the vortex energy of different modes, and accelerated the vortex's spatio-temporal evolution.

**Keywords:** aerodynamic performance; porous media; pore density; vortex structure

## 1. Introduction

Improving the aerodynamics of aircraft has been a continuous goal, and flow control efficiency, whose essence is to trigger local or global flow-field changes by applying physical quantities, such as force, mass, and energy, to the local flow and using the hydrodynamic interactions between the fluids, makes it stand out from other methods [1]. According to whether external energy is required during the control, it is mainly divided into two categories: active or passive flow control [2]. Active flow control techniques are used to improve experimental model performance by directly injecting suitable perturbation patterns into the flow environment. To date, there have been many studies conducted using active flow control techniques to improve aircraft performance [3–5]. However, the system of active flow control is very complex, especially in terms of "robustness" and extra energy consumption. In actual flight, plasma, loop control, and other active flow control technologies that require complex additional equipment have not been applied on a large scale in aircraft, with most remaining in the wind tunnel test stage [6]. Passive flow control does not require auxiliary energy input, and it mainly achieves the purpose of flow control and increasing lift and reducing drag by changing the flow environment, including the boundary conditions, pressure gradient, and other factors. Common passive flow control techniques include leading-edge flaps [7], vortex generators [8], winglets [9], etc. Compared with active flow control, passive flow control is easier to implement and

maintain; therefore, it is more suitable for engineering applications with more complex operating environments. Exploring new flow control technology to enhance the maximum exploitation of the aerodynamic performance potential of aircraft is of considerable academic research value and engineering significance.

Before the Langley Research Center found that a small rib surface could effectively reduce the downstream friction resistance of walls in 1978, it was generally believed that the smoother the surface, the lower the resistance [10]. Researchers have conducted much of research on non-smooth surfaces and have developed many passive drag-reduction control technologies, such as rough wall surfaces [11], superhydrophobic wall surfaces [12], soft skins [13], etc. Inspired by the unique feather structure of "silent flight" owls [14], porous media have been extensively used in the field of drag reduction in recent years. Rasheed et al. carried out a boundary-layer control study on a cone with a half-cone angle of 5.06 degrees under hypersonic conditions, and their experimental results showed that porous media can effectively reduce frictional resistance compared with the experimental results without porous media; furthermore, the conditions with porous media on the wall can even delay the occurrence of transition [15]. Venkataraman and Bottaro performed numerical simulations and found that porous media affect flow topology near the rear of the wing by spontaneously adapting to the separation flow [16]. In 2017, Klausmann and Ruck laid porous media on the downstream side of a cylinder and measured the drag, and their results showed that porous material reduced drag by 13.2% [17]. Joshi and Gujarathi determined that the main cause of porous media drag reduction was the Darcy flow, which transformed the no-slip boundary condition to a quasi-slip Fourier-type and changed the shear force [18]. Li et al. studied an open-channel flow covered by a porous wall with reduced spanwise permeability through direct numerical simulations. They found that the maximum drag-reduction effect of 15.3% occurred when the depth of the porous layer was nine viscous units [19]. Liu et al. summarized that the porous media not only has the effect of noise reduction but also can change the resistance, and by increasing the porosity, the resistance is subsequently reduced. When the porosity is close to 0.97, the highest resistance-reduction effect can be achieved [20].

Mößner et al. installed various porous media in a wind tunnel model and examined them using pressure and PIV measurements. The results show a decrease in lift with increasing permeability. The PIV data show that the airfoil's suction-side boundary-layer thickness increases as a consequence of the flow through the porous trailing edge. Additionally, turbulence increases together with permeability [21]. Aldheeb et al. presented a study on porosity's effect (honeycomb aluminum) on the aerodynamic performance of a symmetric thin airfoil and a straight half-wing based on a thin symmetric airfoil section. They found that drag and moment coefficients change as porosity changes. The drag increases due to increases in viscous stresses caused by flow through the porous region. The lift and lift slope are reduced due to a reduction in the pressure difference between the airfoil's upper and lower surfaces [22]. Thus, different porosity types produce different aerodynamic behaviors. At very low values of porosity, the drag was reduced at low angles of attack. In 2019, Tamaro found that the power spectral density of pressure close to the wall was reduced in the whole frequency spectrum compared with a solid case in correspondence with a porous trailing edge, while the wake structure was highly affected by the presence of porous media [23]. In 2021, Tamaro et al. studied the flow field on solid and porous airfoils (a permeable exoskeleton) subjected to turbulence shed by an upstream cylindrical rod through PIV. Additionally, their flow-field investigation showed that porosity's main effect is to mitigate the turbulent kinetic energy in the stagnation region, attenuating the distortion of turbulence interacting with the airfoil surface [24]. In 2022, Du et al. found that porous media primarily reduced wall resistance because of its "micro-jet" effect, which inhibits the vortex separation frequency and reduces the wall turbulence kinetic energy [25,26]. Although developments in drag-reduction control methods have made considerable progress in porous media research, most studies focus on simple models, such as a flat plate or cylinder, while little research has been conducted on

the wing surface, which is very sensitive to aerodynamic characteristics. The published studies only describe qualitatively that porous media has a certain control effect on wing performance, but the introduction of the influence law on porosity is relatively vague; in particular, its influence mechanisms are still unclear.

Based on this research background, our paper summarizes the effect laws of porous media parameters on the aerodynamic characteristics of airfoils. We also analyze the mechanism of influence on the surface flow-field structure through balance force and pressure measurements and using a TR-PIV experiment that includes laying porous media with different PPI on the airfoil's suction side using a wind tunnel's low-turbulence experimental platform. Our paper's structural framework is as follows: Section 1 introduces the research background of this paper; Section 2 introduces the experimental model and conditions; Section 3 discusses aerodynamic laws; Section 4 analyzes the flow mechanism from the flow-field test results; and Section 5 summarizes the research content and plans future research.

## 2. Experimental Scheme

### 2.1. Wind Tunnel Test System and Experimental Model

The experiment was carried out in the low-turbulence recirculation wind tunnel of the Nanjing University of Aeronautics and Astronautics. The wind tunnel test section measured 3.0 m along the streamwise direction and had a rectangular cross-sectional area of 1.5 m × 1.0 m. The wind tunnel's turbulence intensity, the airflow's deviation angle, and dynamic pressure stability coefficient were less than 0.5%, 0.2°, and 0.05, respectively, and the design's stable wind speed ranged from 5 to 100 m/s. The test section window was equipped with optical glass to provide high-level imaging. The wind tunnel's experimental system, which mainly involves an aerodynamic-balance force measurement system, a pressure testing system, and a TR-PIV system, is shown in Figure 1.

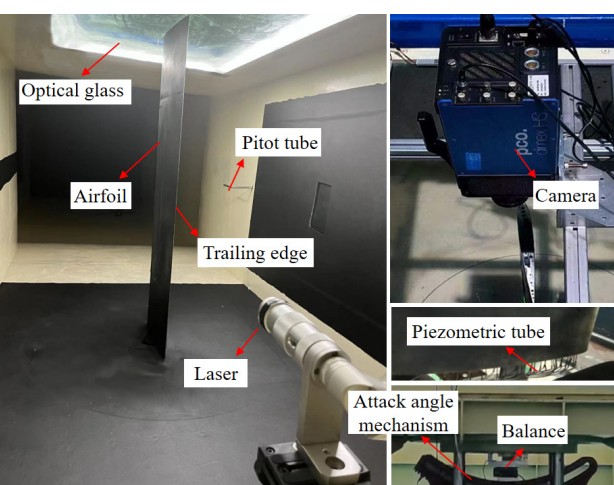

**Figure 1.** Experimental physical layout.

We fixed the airfoil model on the balance using the support mechanism, and then we bolted the balance to the base's attack angle mechanism, while the piezometric tube, which we pre-buried inside the model, penetrated through the round hole on the lower wall of the wind tunnel. We emitted the laser plane parallel to the incoming flow direction from downstream to upstream during the PIV experiment; in this way, we significantly reduced the reflection interference caused by the laser incident on the model's surface. The center of the laser plane coincided with the model's midline, while the camera was fixed at the external upper wall of the wind tunnel, with its lens perpendicular to the laser plane. Moreover, the velocity of the incoming flow was 10 m/s for this experiment, the Reynolds number was $1.37 \times 10^5$, the inlet total pressure was about 101,386 Pa, and the temperature was 288 K.

We adopted the Cartesian coordinate system, and the flow, normal, and spreading direction served as the x, y and z axes, respectively, in which $z = 0$ was denoted as the bottom end of the model. We chose the SD8020 airfoil for our model, which was made of Q235 steel, to ensure strength during the flow. Considering the size of the test section, we designed the chord length and wingspan to be 200 mm and $L = 980$ mm, respectively. In total, there were 25 static pressure holes of 1 mm in outside diameter on the upper and lower surfaces, which we led via a purple copper tube (wall thickness of 0.15 mm, inner diameter of 0.7 mm) from the interior and connected to the pressure-scanning valve by a plastic hose in the terminal. Furthermore, we set 14 and 10 static pressure taps on the upper and lower wing surfaces, respectively, and arranged 1 hole on the leading edge facing the incoming flow, while setting a 5° angle between the holes' direction and the chord to avoid mutual interference. We placed slots at a position 0.2 c from the leading edge, where the maximum depth and size were 10 mm and 40 mm × 980 mm, respectively. We laid same-size porous media in the slot to ensure a consistent shape with the airfoil's outer profile. The model design is shown in Figure 2, and Table 1 shows the pressure taps' pressure distributions.

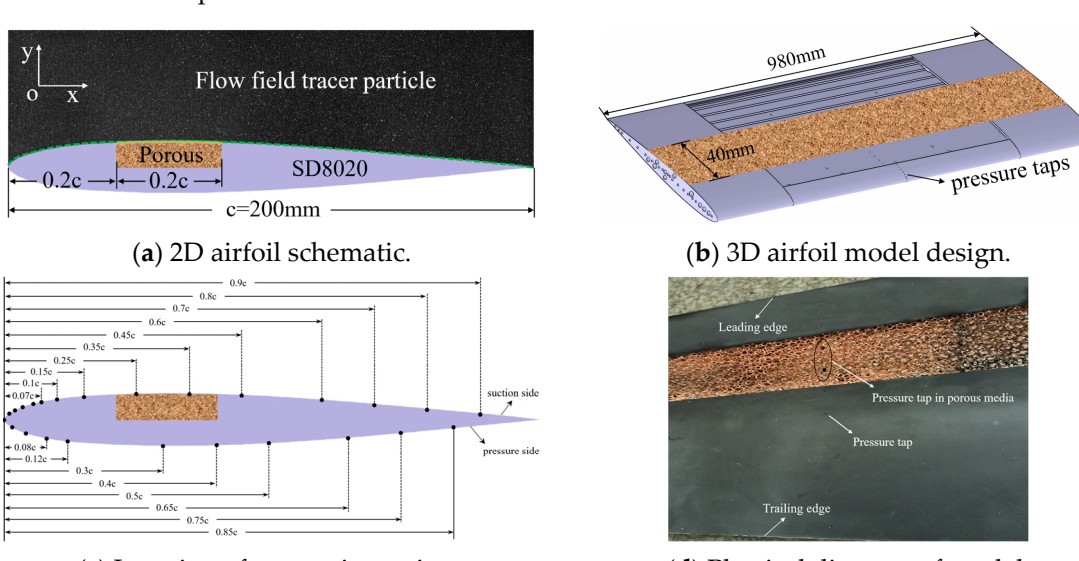

(**a**) 2D airfoil schematic.

(**b**) 3D airfoil model design.

(**c**) Location of measuring points.

(**d**) Physical diagram of model.

**Figure 2.** Model design schematic diagram.

**Table 1.** Distribution of pressure taps on airfoil surface.

| Number | | 1 | 2 | 3 | 4 | 5 | 6 | 7 | 8 | 9 | 10 | 11 | 12 | 13 | 14 |
|---|---|---|---|---|---|---|---|---|---|---|---|---|---|---|---|
| Suction side | $x/c$ | 0.01 | 0.02 | 0.035 | 0.055 | 0.07 | 0.1 | 0.15 | 0.25 | 0.35 | 0.45 | 0.6 | 0.7 | 0.8 | 0.9 |
| | $z/L$ | 0.5002 | 0.5004 | 0.5006 | 0.501 | 0.5012 | 0.5018 | 0.5027 | 0.5045 | 0.5062 | 0.508 | 0.5107 | 0.5227 | 0.5245 | 0.5263 |
| Pressure side | $x/c$ | 0.015 | 0.04 | 0.08 | 0.12 | 0.3 | 0.4 | 0.5 | 0.65 | 0.75 | 0.85 | / | / | / | / |
| | $z/L$ | 0.5003 | 0.5007 | 0.5014 | 0.5021 | 0.5034 | 0.5071 | 0.5089 | 0.5116 | 0.5234 | 0.5252 | | | | |

We used copper foam as the porous media, which is a multifunctional material with a large number of connected or unconnected holes uniformly distributed in the copper matrix and is commonly used for thermal conductivity, pressure buffering, and noise reduction. This material can stabilize vortices, change wall structures, and has excellent drag-reduction performance due to its unique inner structure. In addition, the copper foam material is soft and easy to cut, ensuring a perfect fit with the airfoil's surface. We investigated three copper foam porous pads with 5 PPI, 20 PPI, and 60 PPI, whose permeabilities were $6.27 \times 10^{-7}$ mm², $1.31 \times 10^{-7}$ mm², and $1.24 \times 10^{-8}$ mm², respectively, with 98% porosity in all cases. Images of the porous media are shown in Figure 3. The three selected porous media had large pore density spans and considerable permeability changes; therefore, we could obtain the general control rule through our experiment.



(**a**) 5 PPI.　　　　　　　(**b**) 20 PPI.　　　　　　　(**c**) 60 PPI.

**Figure 3.** Images of porous media.

### 2.2. Experimental Equipment and Test Methods

The force measurement experiment used a six-component cassette strain balance, which separately measured six components (X, Y, Z, Mx, My, and Mz) through mechanical decomposition with a bridge circuit. The balance response frequency was greater than 80 Hz, the load range of the balance in the z direction was 50 kg, the load range in x direction was 10 kg of the resistance direction, the load range in y direction was 15 kg of the lift direction, and the calibration accuracy and precision were 0.28% and 0.041%, respectively, fully meeting the requirements of our experiment. In the formal trial, the sampling frequency of the force measurement was 2000 Hz with a sampling time of 5 s, which means we obtained the final force data from 10,000 points of time-averaged results. The airfoil lift and drag coefficient can be calculated by Equations (1) and (2), respectively.

$$C_L = \frac{F}{0.5\rho U^2 SL} \tag{1}$$

$$C_D = \frac{D}{0.5\rho U^2 SL} \tag{2}$$

where $U$ is the incoming flow velocity, $S$ is the projected area of the airfoil, $L$ is the spread length of the airfoil, and $F$, $D$ are the total lift and drag forces gauged by the balance, respectively.

In the experiment, we used two DSA3217-PTP/16Px with 16-channel pressure-scanning valves to gauge the static pressure on the airfoil's surface; moreover, the pressure-scan valve's sampling frequency was 1000 Hz, with a sampling time of 5 s. The pressure-scanning valve we used had a water-column-pressure measurement range of 10 inches, which is approximately 2540 Pa, with an accuracy within 0.05% of the full scale. The surface pressure coefficient ($C_P$) distribution can be calculated by Equation (3).

$$C_P = \frac{P}{0.5\rho U^2} \tag{3}$$

where $P$ is the static pressure measured by the pressure-scanning valve.

We used the TR-PIV to test the flow field on the airfoil's suction side, which consists of a double-pulse laser (Vlite-Hi-527-30, divergence angle $\leq$ 5 mrad), a PCO high-speed camera (2000 pixel $\times$ 2000 pixel, 12 bit), a synchronization controller, and a smoke generator. Before the experiment, we sprayed a layer of black matte paint with a thickness of approximately 20 μm on the wing model's surface to absorb the laser and further eliminate reflection issues. In the PIV experiment, we set the laser's luminescence frequency of to 1000 Hz, the pulse width of both A and B lasers to 50 μs, and the interval time to 60 μs. The laser plane's intensity was uniform in the flow-field test area, with a thickness of approximately 1 mm. We equipped the camera lens with a 750 nm high-pass filter, set the camera's shooting frame rate to 1000 fps, activated a double exposure mode, and assigned an 80 μs exposure time, while controlling the laser's light source and the camera's time sequence using a synchronization controller to filter out the excitation and ambient light, in addition to other sources of stray light interference. The wind tunnel had two experimental sections. Because the wind tunnel is backflowing, we placed the smoke generator upstream of the model (at another test section) to spray tracer particles through the wind tunnel observation

window to observe the tracer particle situation, after which we uniformly distributed the concentration to start recording the flow-field picture. The trace particles, which were generated from smoke-oil vaporing at sizes of 1μm, displayed high followability and had negligible impact on the flow field. Before post-processing, we filtered 1000 pairs of transient raw images using the MATLAB filtering program, and then we used the PIV-view software to perform correlation analysis on the processed images. We used the standard FFT correlation algorithm with an interrogation window of 32 pixel × 32 pixel and an overlap factor of 50% to determine the correlation and thus obtain the particle displacement.

### 2.3. Ω Vortex Identification Method

The vorticity in the flow field is defined as the spin of the velocity, which is of great value in flows dominated by vortex structures. However, the magnitude of vorticity is not able to precisely represent the existence of an actual vortex. For example, there are vortices in the laminar boundary layer, but no rotational motion actually occurs. Liu et al. [27] proposed a method for vortex identification: the vortex volume is further decomposed into a spin and spinless part with the introduction of the parameter Ω, which is to define and identify the vortex, as shown in Equation (4).

$$\Omega = \frac{1}{2}\left[\frac{\left(\frac{\partial u}{\partial y}\right)^2 + \left(\frac{\partial v}{\partial x}\right)^2 - \left(\frac{\partial v}{\partial x}\right)\left(\frac{\partial u}{\partial y}\right) - \left(\frac{\partial u}{\partial x}\right)^2}{0.01 + \left(\frac{\partial u}{\partial y}\right)^2 + \left(\frac{\partial v}{\partial y}\right)^2 + \left(\frac{\partial v}{\partial x}\right)^2 - \left(\frac{\partial u}{\partial x}\right)^2}\right] \tag{4}$$

where $\Omega \in [0, 1]$ represents the ratio of the vortex vorticity of the total value, and the flow can be classified into translational, deformation, rotational, and non-rotational motions according to this way of decomposition. $\Omega = 0$ represents pure deformation motion, and $\Omega = 1$ means pure rotational motion. Based on the theory of Liu et al. [28], the threshold value of Ω is generally taken as 0.52, and the vortex core boundary selection of the threshold $\Omega \geq 0.95$.

## 3. Aerodynamic Measurement Results and Discussion

### 3.1. Aerodynamic Force Results

In this experiment, lift force accounted for approximately 8% of the scale in the y direction of the balance, and drag force accounted for approximately 3.5% of the scale in the x direction, meaning both forces conformed to the measurement requirements. Figure 4 shows the smooth airfoil's force measurement results by repeated balance three times. The figure shows that the three measurements are basically consistent, indicating that the balance has high repeatability.

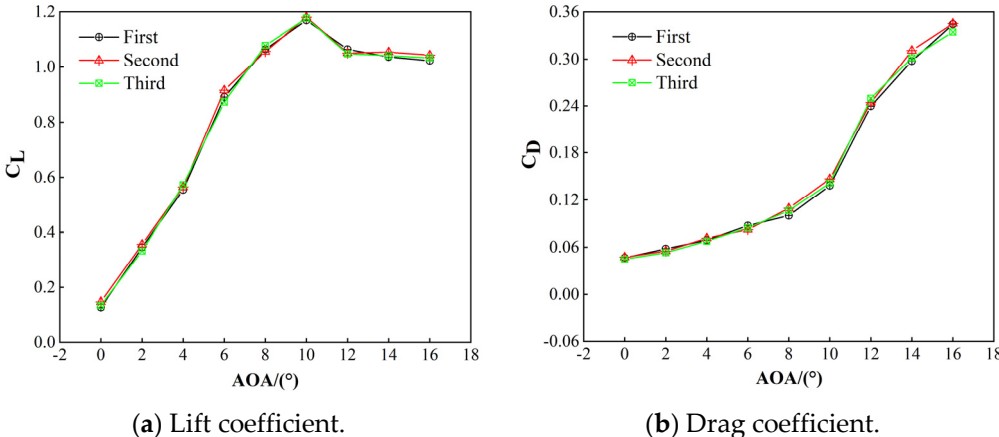

(**a**) Lift coefficient.　　　　　　　　　　　　　(**b**) Drag coefficient.

**Figure 4.** Verification of force repeatability of balance.

From Figure 5a, we can see that the lift coefficient decreases after laying with porous media, and the decrease becomes more obvious with the increase in PPI. The lift coefficient's rate decreases more rapidly in the range of small angles of attack and slows down when the angle of attack is larger than 10°. The drag distribution curve in Figure 5b shows that the drag coefficients of the wings laid with 60 PPI porous media increase, while the drag coefficients with 5 PPI and 20 PPI porous media increase and significantly decrease respectively, and with 20PPI porous media they have the highest drag-reduction performance.

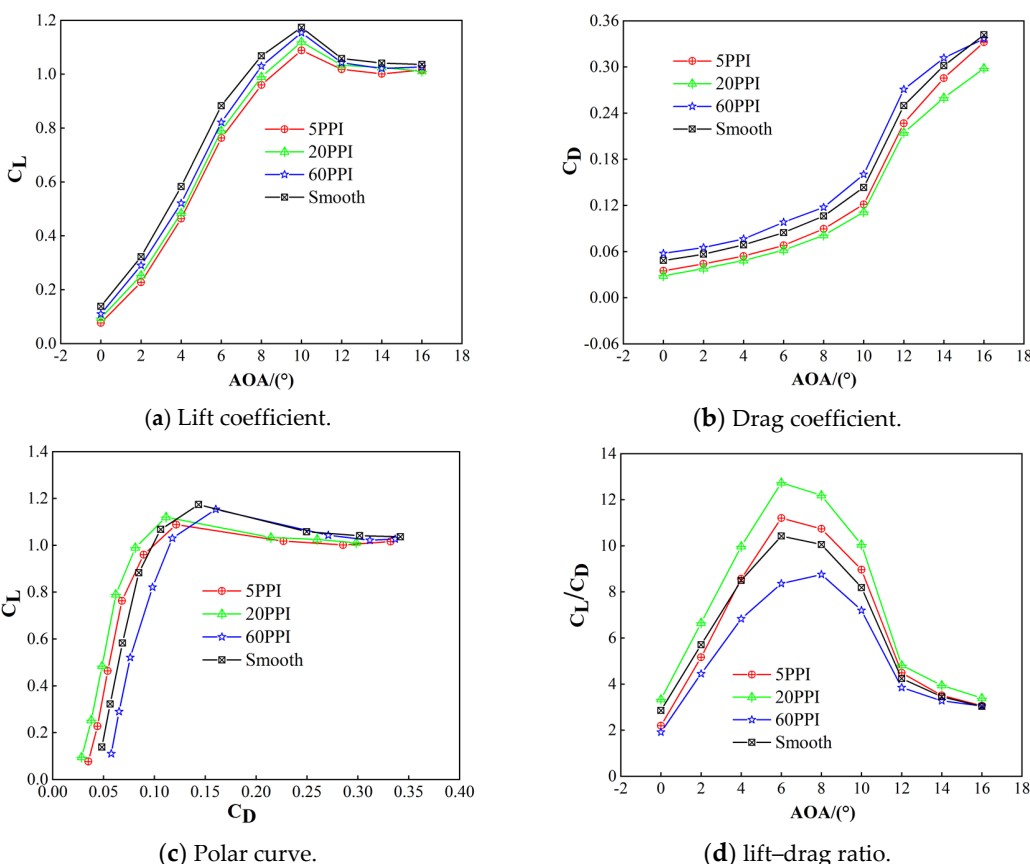

(**a**) Lift coefficient.

(**b**) Drag coefficient.

(**c**) Polar curve.

(**d**) lift–drag ratio.

**Figure 5.** Aerodynamic comparison of airfoil–laying porous media of different pore densities.

Figure 5c shows the that the pole curve of the airfoil with 20 PPI porous media shifted significantly to the left compared with the smooth airfoil, indicating that the aerodynamic performance considerably improved before the angle of attack reached 10°. For further comparison, the lift–drag ratio's characteristic curve in Figure 5d shows that 60 PPI porous media seriously damaged aerodynamic performance, with a 18.2% reduction in the maximum lift–drag ratio. When laid with 5 PPI porous media, the airfoil showed a slight decrease and increase in lift–drag ratio before and after 4°, respectively. The airfoil with 20 PPI porous media showed a considerable increase in lift–drag ratio, and the maximum lift–drag ratio increased by 27.4% at a 6° angle of attack. For this condition, we conducted three repetitions of related experiments, which have reasonably consistent data. We carried out further experimental studies to explain this phenomenon.

## 3.2. Pressure Distribution Results

According to the balance force measurement results, we found that the airfoil had the largest lift–drag ratio at a 6° angle of attack, and the control effect of porous media was most considerable at 6°, while after the stalled angle of attack, the porous media still had a large effect on the airfoil's aerodynamic performance. To simplify the experiment, we conducted the pressure measurement test at a 6° angle of attack corresponding to the maximum lift–drag ratio and the 12° angle of attack of the wing's deep stall.

Figure 6 shows the characteristic curves of the airfoil's surface pressure distribution after being laid with porous media of varying pore density (6° and 12° angle of attack with 10 m/s incoming flow velocity). Figure 6 highlights that the airfoil's side pressure distribution does not change greatly after laying porous media, while the airfoil's suction side negative pressure gradually decreases with the reduction in PPI. The negative pressure is considerably reduced, particularly in the porous media area, thereby causing the decrease in lift coefficient, which is consistent with previously mentioned force measurement results. This result may be due to reductions in the porous media's PPI. When the material's pore diameter becomes larger, more airflow may enter the interior of the porous media and hinder airflow movement, thus causing the negative pressure to decrease.

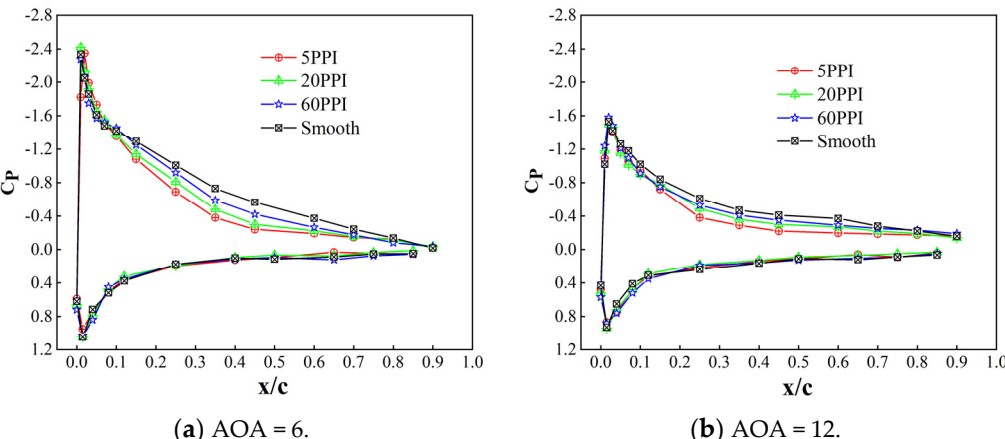

(**a**) AOA = 6.　　　　　　　　　　　　　　　　(**b**) AOA = 12.

**Figure 6.** Pressure distribution of airfoil-laying porous media of different pore densities.

The inverse pressure gradient increases near the porous media area compared with the smooth airfoil, while the force results show that the total drag decreases with the 5 PPI and 20 PPI porous media, indicating that this porous media increases pressure drag to some extent; however, it substantially reduces frictional drag. Because pressure drag accounts for a relatively small proportion of the total drag under the test conditions, the frictional drag comprises the majority of the drag. Therefore, it is not contradictory that the total drag declines while the pressure drag increases.

## 4. Flow-Field Results and Discussion

Through force and pressure analysis, we found that the porous media have a considerable effect on the airfoil's aerodynamic performance in the entire AOA test range. Therefore, in order to improve efficiency during the following experiment, we decided to select two typical AOA for the PIV test (a small and a large AOA). The main reason for choosing the 6° AOA was that the lift–drag ratio reached its peak in this situation, while the porous media's control effect continued to perform at a high level. Furthermore, we chose the 12° AOA mainly because of our aim to explore the control mechanisms of porous media on airfoil aerodynamics after stall, while the flow structure was obviously revealed in the separation zone under this AOA, thus making it convenient for comparison.

In Section 4.1, we first discuss the effect of pore densities on time-averaged velocity and shear stress fields. Subsequently, we undertake an analysis of the unsteady flow field in Section 4.2. For the sake of simplification, we examine the development law of a vortex under the highest control condition compared with the smooth airfoil, which we laid with porous media of 20 PPI and while positioning a 12° angle of attack. Finally, in Section 4.3, we analyze the vortex's spatial–temporal evolution processes of the transient flow field using the dynamic mode-decomposition method (DMD) to analyze the mechanisms inherent in the changes in the aerodynamic properties of porous media.

### 4.1. Time-Averaged Flow Field

4.1.1. Time-Averaged Velocity Field

The time-averaged velocity field describes the speed of airflow movement in a period of time, which reflects the overall flow state and facilitates the visual determination of flow characteristics. It can be obtained by averaging the transient velocities in a region over a period of time.

$$\overline{u} = \frac{1}{N} \sum_{i=1}^{N} u_i \tag{5}$$

$$\overline{v} = \frac{1}{N} \sum_{i=1}^{N} v_i \tag{6}$$

$$velmag = \sqrt[2]{\overline{u}^2 + \overline{v}^2} \tag{7}$$

where $N$ is the number of sampling points in the measurement time $t$; $\overline{u}$ and $\overline{v}$ are acquired from the PIV experiment data.

The time-averaged velocity field contours of the suction side are given in Figures 7 and 8, which show models laid with different PPI values of porous media while under the 6° and 12° angles of attack. From Figure 7, we see that the flow velocity of the airfoil laid with porous media shows a decreasing trend at a small angle (AOA = 6°) compared with the smooth airfoil. As the PPI value decreases, the low-speed area at the tail increases considerably. For example, when using 5 PPI, the trailing velocity drops considerably, indicating that the porous media somewhat hinder the airflow movement, thus reducing the circumfluence circulation and leading to reduction in the lift coefficient. From Figure 8, we observed that the airfoil's suction side separation region is expanded at a large angle of attack (AOA = 12°), likely due to the comparatively rough surface that characterizes porous media. As the PPI value reduces, the separation region gradually increases. Meanwhile, the vortex size in the separation region when laying 60 PPI porous media increases slightly compared with that of the smooth airfoil, while the separation vortex size when laying 20 PPI porous media greatly decreases. This indicates that although 20 PPI porous media has a certain loss of lift, it can weaken the vortex energy and inhibit large-scale vortices to a certain extent, thus improving the airfoil's aerodynamic performance.

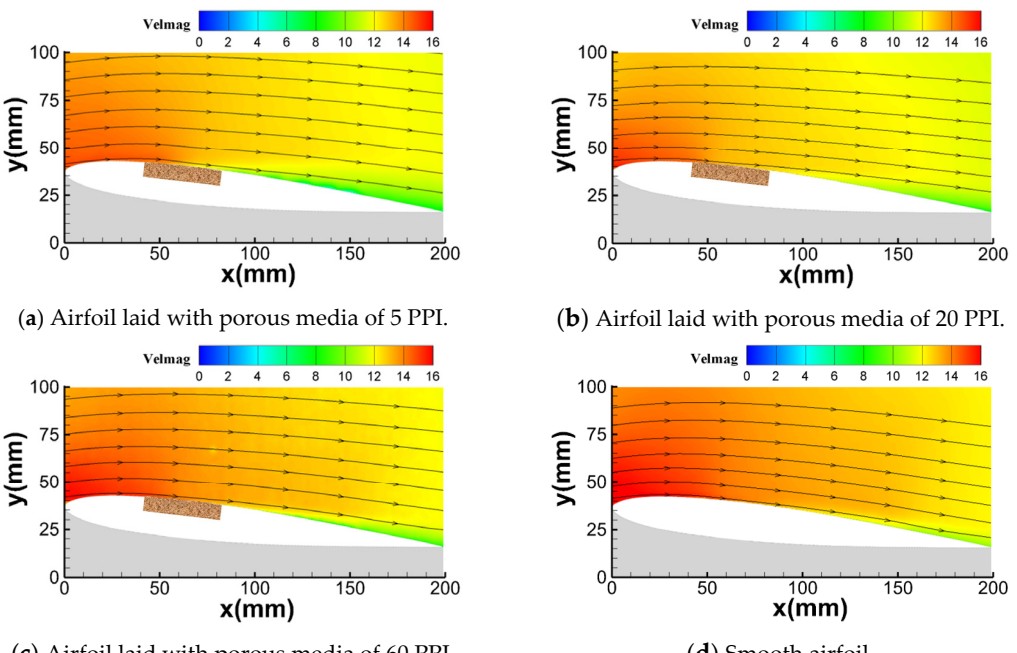

(**a**) Airfoil laid with porous media of 5 PPI.

(**b**) Airfoil laid with porous media of 20 PPI.

(**c**) Airfoil laid with porous media of 60 PPI.

(**d**) Smooth airfoil.

**Figure 7.** Time-averaged results of velocity field (AOA = 6).

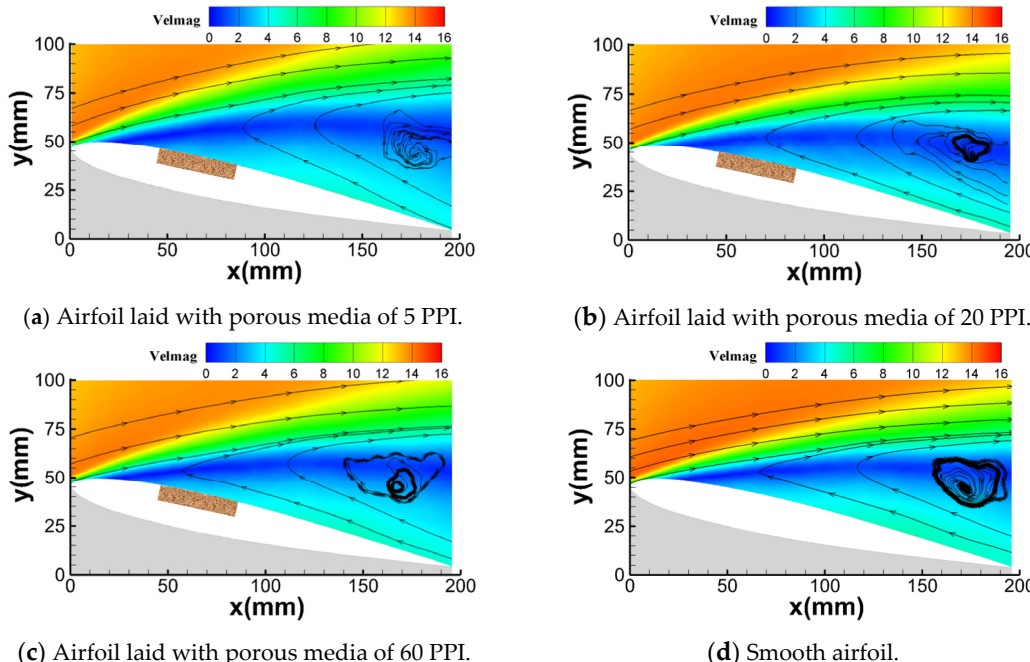

(**a**) Airfoil laid with porous media of 5 PPI.

(**b**) Airfoil laid with porous media of 20 PPI.

(**c**) Airfoil laid with porous media of 60 PPI.

(**d**) Smooth airfoil.

**Figure 8.** Time-averaged results of velocity field (AOA = 12).

We captured and identifies the vortex characteristics within the separation region on the suction by means of the $\Omega$ vortex identification method mentioned above. It can be seen from Figure 9 that the boundary of the vortex core changes after laying the porous media. Compared with the smooth airfoil, the boundary of the vortex core in the airfoil separation area with 60 PPI porous media increases, which may lead to the increase of friction resistance. But the boundary of the vortex core in the airfoil separation area with 5 PPI and 20 PPI porous media is significantly reduced, and the effect caused by 20 PPI is the most significant, greatly improving the aerodynamics. This conclusion is consistent with Figure 8.

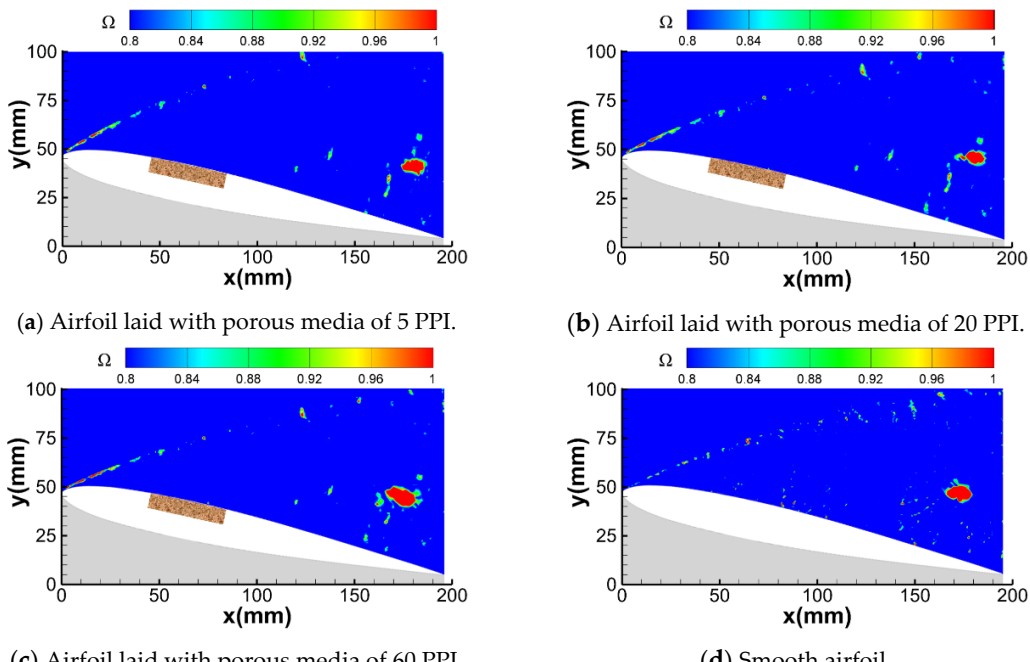

(**a**) Airfoil laid with porous media of 5 PPI.

(**b**) Airfoil laid with porous media of 20 PPI.

(**c**) Airfoil laid with porous media of 60 PPI.

(**d**) Smooth airfoil.

**Figure 9.** Time-averaged results of vortex core boundary with $\Omega$ criterion (AOA = 12).

### 4.1.2. Time-Averaged Shear Stress Field

Figure 10 presents the time-averaged shear stress contours at the 6° angle of attack, which consists of the results of 5 PPI, 20 PPI, and 60 PPI porous media laid on both the airfoil and smooth airfoil. Possibly due to the existence of experimental interference factors, the flow field on the test model's suction side was not completely attached at the 6° angle of attack and still had a small range of separation. As seen from the figure, the surface's shear layer is thickened due to the existence of porous media, which suppresses the airflow's velocity on the upper airfoil. However, although the shear layer of the airfoil laid with 20 PPI porous media is thicker than that of the smooth airfoil, the wall shear force is considerably lower, thus reducing the wall friction and improving aerodynamic performance. The reason may be that the size of the 20 PPI porous media pore is similar to the wall's spanwise vortex, which leads the spanwise vortex to be "locked" in the porous media pore. Because the rolling friction is much less than the sliding friction, other vortices flow smoothly across the wall, reducing the momentum exchange of low/high speed fluid inside/outside the shear layer, thus reducing the shear stress between the fluid and airfoil.

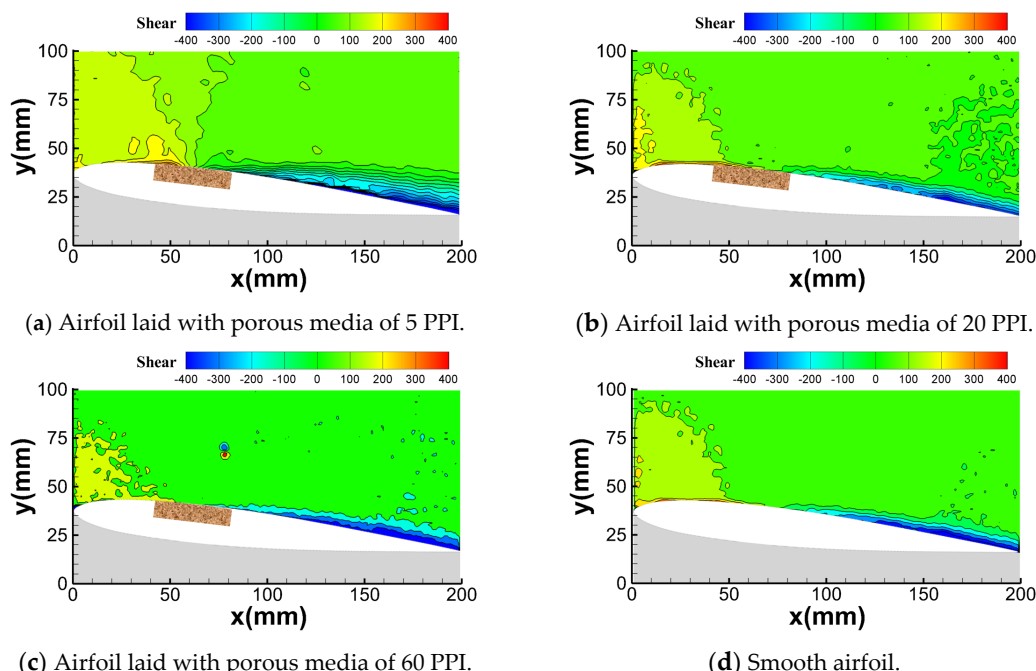

(**a**) Airfoil laid with porous media of 5 PPI.

(**b**) Airfoil laid with porous media of 20 PPI.

(**c**) Airfoil laid with porous media of 60 PPI.

(**d**) Smooth airfoil.

**Figure 10.** Time-averaged results of shear stress field (AOA = 6).

### 4.2. Unsteady Flow Field

We selected a 12° AOA for unsteady flow-field analysis because the airfoil's flow-field structure is not obvious and is difficult to capture at a small angle of attack, while at a 12° AOA, large flow separation occurs, and the vortex structure in the separation area is obvious, which is convenient for comparison. The second reason is that we also want to study the control mechanisms of porous media on airfoil aerodynamic performance and the spatio-temporal evolution law of vortices at a deep-stall angle of attack.

Figure 11 gives three consecutive instantaneous vortex contours based on the $\Omega$-vortex identification criterion at 12° angles of attack for the airfoil laid with 20 PPI porous media and the clean airfoil. From the figure, we observe that the number of vortices increased on the suction side laid with 20 PPI porous media, the vortex size tended to be finer, and the $\Omega$ value was reduced compared with the clean airfoil, indicating that the proportion of rotating vortices in this region was relatively small. Meanwhile, porous media can break the large-scale high-energy vortex into more small-scale vortices of low energy, accelerating vortex dissipation and achieving improvements in aerodynamic characteristics.

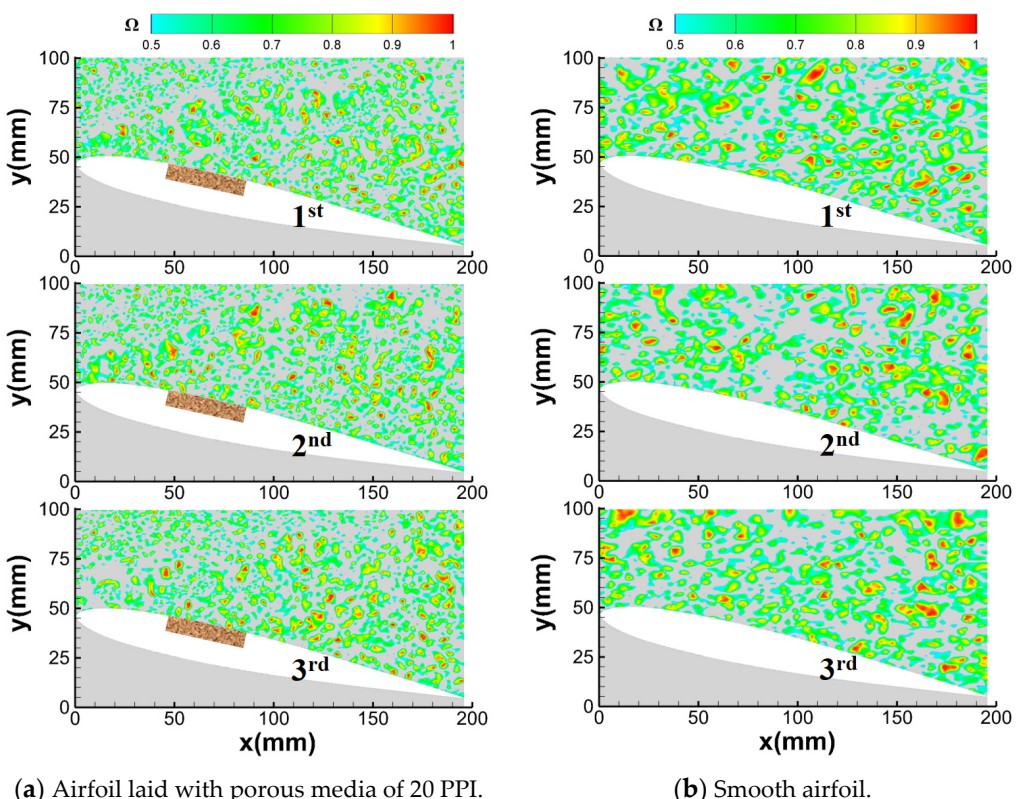

(**a**) Airfoil laid with porous media of 20 PPI.

(**b**) Smooth airfoil.

**Figure 11.** Vorticity contours with Ω criterion.

### 4.3. DMD Mode Analysis

The DMD method is a decomposition technique that can reduce the dimensionality of the dynamic system. The accurate identification of the vortex by modal step-down can capture the vortex frequency, wavelength, and propagation velocity of each mode, and tracking the development and evolution of the vortex structure can result in more information about the dynamics of the original system. Each mode obtained with the DMD method has a unique growth rate and frequency magnitude, which facilitate the analysis of complex flows [29–31].

### 4.3.1. Mode Distribution

Modes can be classified according to their types, such as quasi-static, drifting, and conjugate modes, where the quasi-static mode, also called the first order mode, is similar to the average flow field. Table 2 gives the first 15 order modes of the airfoil's suction side vortex field for the airfoil laid with porous media of 20 PPI and the smooth airfoil at an angle of attack of 12°, classified according to the type of mode, noting that the modes under each type are ordered by energy magnitude.

### 4.3.2. Vortex–Mode Energy

Figure 12 shows the relative energy distribution of the first 15 order modes of the vorticity field on the suction side of the airfoil laid with porous media of 20 PPI and the smooth airfoil at an angle of attack of 12°. From the figure, we observed that the first six order modes' energy accounts for approximately 70% of the total energy, which indicates that the DMD method used in this test is consistent with the physical characteristics, and its reduced-order reconstruction results are better. In comparison, we found that the airfoil's first-order mode energy when laid with porous media was less than that of the smooth airfoil, and the first-order modal energy was similar to the average flow field, indicating that porous media can reduce the vortex energy of the average vortex field on the upper airfoil. At the same time, we found that the slope of the first nine order mode curves of

the airfoil laid with porous media is considerably larger than that of the smooth airfoil, indicating that the porous media accelerates the main modal vortex dissipation rate.

**Table 2.** First 15 DMD mode classification.

| Mode Type | Airfoil Laid with Porous Media | Smooth Airfoil |
|---|---|---|
| Quasi-static | 1st | 1st |
| Drifting | 4th<br>7th<br>12th<br>15th<br>/<br>/ | 2nd<br>7th<br>10th<br>13th<br>14th<br>15th |
| Conjugate | 2nd and 3rd<br>5th and 6th<br>8th and 9th<br>10th and11th<br>13th and 14th | 3rd and 4th<br>5th and 6th<br>8th and 9th<br>11th and 12th<br>/ |

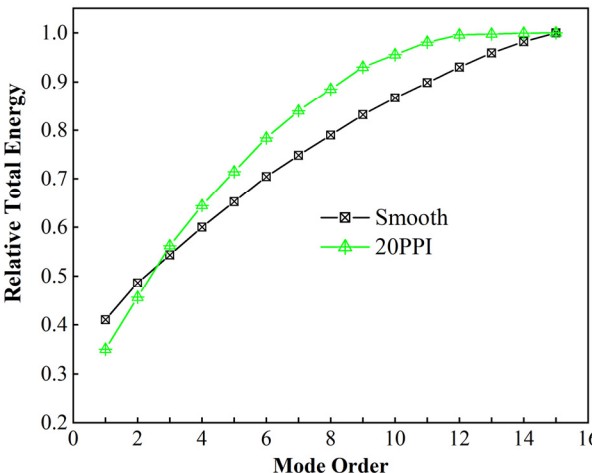

**Figure 12.** Relative energy distribution of modal order.

Figure 13 provides the DMD amplitude and frequency distribution of the first 15 order modes of the vortex field on the smooth airfoil and the airfoil's suction side with a 12° angle of attack for the airfoil laid with 20 PPI porous media. Because the quasi-static mode's flow field does not change with time, it does not grow or decay, and its frequency is zero, while the drifting mode eigenvalue is zero in the imaginary part; therefore, its frequency is also zero. From the figure, we observed that the DMD amplitude of the first-order mode of the airfoil laid with porous media was considerably lower than that of the smooth airfoil, which corresponds to the results in Figure 12. We also discovered that the conjugate mode appeared earlier after laying the porous media compared with the smooth airfoil, indicating that the porous media sped up the mode evolution process.

Comparing the mode frequencies, we found that the frequency increased with the increase in the mode order, indicating that the vortex energy gradually dissipated, and that the low-order-mode vortex had a large scale, high energy, and low frequency, while the high-order-mode vortex had a small size, low energy, and high frequency. After laying the porous media, the mode frequency considerably increased compared with the smooth airfoil, which indicated that the porous media can break the large-scale vortex structure into more small-scale vortex structures, effectively weakening the energy of different modes and at the same time making the high-order modes stabilize more quickly by absorbing high-frequency disturbances, accelerating the vortex evolution process, and thus improving the airfoil's aerodynamic performance.

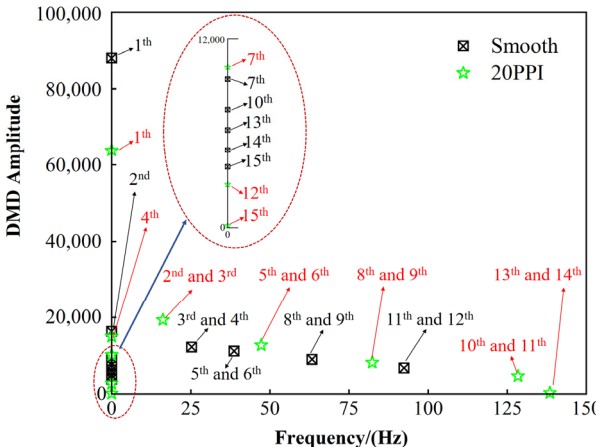

**Figure 13.** Amplitude and frequency distribution of different mode orders.

### 4.3.3. Conjugate Mode of the Vorticity Field

Figure 14 shows the first three pairs of conjugate modes of the airfoil's suction side vortex volume field modal amplitude ordering for the airfoil laid with 20 PPI porous media and the smooth airfoil at the 12° angle of attack. Only one pair of conjugate modes is shown in the figure because the structure of the pair of conjugate modal flow fields is the same.

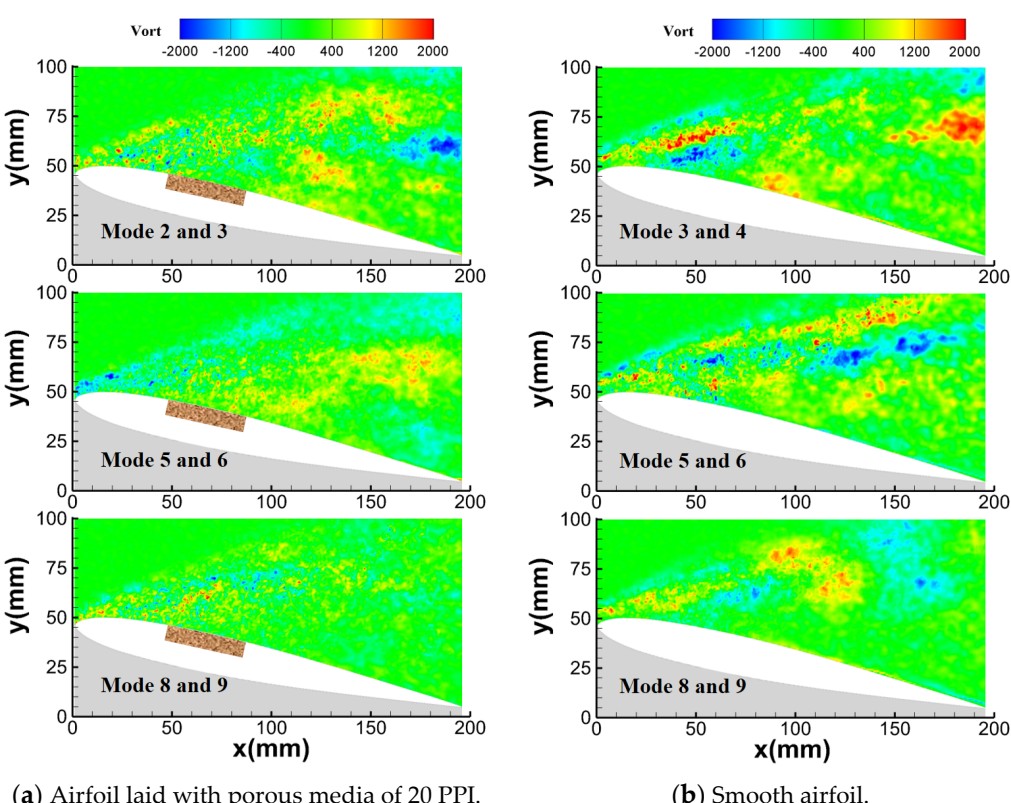

(**a**) Airfoil laid with porous media of 20 PPI.  (**b**) Smooth airfoil.

**Figure 14.** Conjugate modes.

We found that the vorticity gradually decreased with the increase in the mode order, indicating that the vortex gradually lost energy during the evolution process. Comparing the results in Figure 14, we discerned that the vorticity values of the corresponding modes were lower, with the vortex size relatively smaller after laying with the porous media, indicating that the vortex strength was weaker. At the eighth and ninth modes, the smooth airfoil's vorticity was obviously distinguishable, while the vorticity of the airfoil laid with

porous media was almost dissipated, further indicating that the porous media promoted the vortex evolution process.

## 5. Conclusions

In this paper, we carried out an experimental study on the effect of porous media on the aerodynamic performance of airfoils by combining three test methods: force measurement by pneumatic balance, pressure distribution measurement by pressure-scanning valves, and flow-field testing by TP-PIV. The main conclusions are as follows:

(1) Only the porous media with the appropriate pore density (20 PPI) could significantly improve the aerodynamic performance of the airfoil. If the pore density of the porous media is too small, the aerodynamic performance of the airfoil will be seriously damaged in the whole range of the angle of attack. If the pore density is too large, the porous media may act like a spoiler, increasing the viscous effect, and the aerodynamic power of the airfoil will be reduced under the condition of a small angle of attack;

(2) Porous media (20 PPI) mainly reduce the drag by considerably reducing the airfoil surface's frictional resistance while the pressure resistance increases. It also can weaken the wall shear stress.

(3) Porous media (20 PPI) can destroy the vortex structure, breaking a large-scale vortex with low-frequency into a high-frequency granular vortex, inhibit the amplitude of vortex fluctuation, effectively weaken the energy of different modes of the vortex, accelerate the vortex evolution process, and thus improve the airfoil's aerodynamic performance.

In this paper, we obtained the law and mechanism influences of porous media with different pore densities on the aerodynamic performance of airfoils, and the corresponding results can provide technical support in the field of aerospace passive drag reduction. Future work will consider the flow control of porous media under extreme conditions, such as an ultra-high Reynolds number and a deep cryogenic temperature environments.

**Author Contributions:** W.K.: Data curation (lead); Formal analysis (lead); Methodology (lead); Software (lead); Writing—original draft (lead). H.D.: Conceptualization (lead); Writing—review & editing (lead). J.W.: Writing—review & editing (equal). Y.Z.: Investigation (equal); Visualization (equal). Z.J.: Investigation (equal); Validation (equal). All authors have read and agreed to the published version of the manuscript.

**Funding:** This research was funded by the National Natural Science Foundation of China (No. 11872208), the National Numerical Wind Tunnel Project of China (Grant No. 0747-2266SCCMY003), and the Fundamental Research Funds for the Central Universities (No. NF2020001).

**Institutional Review Board Statement:** Not applicable.

**Informed Consent Statement:** Not applicable.

**Data Availability Statement:** The data that support the findings of this study are available from the corresponding author upon reasonable request.

**Conflicts of Interest:** The authors declare that they have no known competing financial interest or personal relationships that could have appeared to influence the work reported in this paper.

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
