# Peer review of "Experimental Study on the Effect of Porous Media on the Aerodynamic Performance of Airfoils"

_aerospace, doi:10.3390/aerospace10010025_

Round 1
Reviewer 1 Report
The authors have experimentally investigated the effect of porous media on
aerodynamic performance of airfoil. The findings are very interesting. My suggestions are given below:
1) Include more information on the PIV experiment procedure.
2) Is the Reynolds number laminar or turbulent?
3) In section 3.1 authors have described time-averaged velocity field. How the averaging is performed?
4) The authors should discuss the reason for choosing 6 and 12 degree AOA.
Author Response
Dear Reviewer:
Thank you very much for your comments on our manuscript. We have carefully read all your comments and replied point-by-point. Please see the attachment for details.

Reviewer 2 Report
The manuscript presents an experimental campaign describing the influence of porous media distributed over the wing on airfoil performance. The authors compare the results of various pore densities in terms of pressure distributions and flow field distribution thanks to wall pressure and Particle Image Velocimetry measurements.
In my view, the results presented by the authors are detailed and relevant to the inclusion of porous media on airfoil surfaces. The revealed performance influence of the porous media would help the future integration or development of porous sections over different airframe surfaces. However, the current description of the methodology, description of results, and conclusions limit the outreach of the presented research. In this line, before recommending its publication I suggest the authors address the following items to improve the methodology description and the outreach of the results:
• The authors should include more references on the use of porous media towards increase airfoil performance and describe the lessons learned and open challenges raised in those investigations that could further support the need for the presented experimental campaign
· When describing the experimental set-up the authors should include the test section inlet total pressure and temperature during the experiments
· Is the Reynolds number based on the airfoil cord?
· Instead of using upper or lower wing surfaces, please describe them as suction side or pressure side.
· Describe in further detail the distribution of the measurement points, it would be valuable to include a figure indicating the measurement locations.
· Include in table 1 also the span-wise location of the pressure taps
· Why did the authors select that position for the porous media inclusion? Please explain why is that position relevant not any other along the airfoil
· Why was copper foam selected as the porous media? Please justify its selection
· Why were those pore densities selected? Please justify their selection
· Figures 4 and 5 do not add real value to the manuscript I suggest removing them.
· Why is the sampling time limited to 5 s?
· The results being presented are then a temporal average of those 5 seconds of sampled data.
· What was the sampling frequency of the pressure measurements?
· What was the distance between the pressure taps and the pressure scanner? That distance affects the actual frequency resolution of the measurements.
· Describe explicitly the actual pressure resolution of the pressure measurements and its relative size to the actual pressure variations measured.
· Some balance details given in 2.1 are redundant,
· Include an analysis of the lift and drag coefficients uncertainty. Reflect that uncertainty on the figure results
· Remove the linear evolution between the discrete measurement points in Figures 6 and 7. Only the measured points are real, the linear imposed distribution between points may drive erroneous conclusions.
· When analyzing the Cd/Cl results describe the mechanisms that may be leading towards that behavior.
· Why are the presented angles of attack in Figure 7 relevant and not the ones of any other AoA?
· Explain why the 12 degrees AOA is selected for the analysis presented in section 3.2
· Include some quantitative measurements of the size of the recirculation when describing Figure 9 results.
· Describe what could be the mechanism driving the recirculated flow region performance for the different pore densities studied.
Author Response

(The authors gave the same response as above.)

Round 2
Reviewer 2 Report
The manuscript presents an experimental campaign describing the influence of porous media distributed over the wing on airfoil performance. The authors compare the results of various pore densities in terms of pressure distributions and flow field distribution thanks to wall pressure and Particle Image Velocimetry measurements.
In my view, the results presented by the authors are detailed and relevant to the inclusion of porous media on airfoil surfaces. The revealed performance influence of the porous media would help the future integration or development of porous sections over different airframe surfaces. The authors have considerably improved both the introduction and description of methodology. In addition, the manuscript presents now a more justified case of study and broader conclusions can be drawn from the outlined results. In this line, I recommend this manuscript for publication
Author Response
Dear Reviewer:
Thanks for your approval. We really appreiate your sincere suggestions,which has given us invaluable help to improve our paper, and will still greatly guide our future work.
Best wishes!
Hao Dong